# Rapid task-dependent tuning of the mouse olfactory bulb

**Anzhelika Koldaeva[1], Andreas T Schaefer[2,3]\*, Izumi Fukunaga[1,2]\***

[1]Sensory and Behavioural Neuroscience Unit, Okinawa Institute of Science and Technology Graduate University, Okinawa, Japan; [2]Neurophysiology of Behaviour Laboratory, The Francis Crick Institute, London, United Kingdom; [3]Department of Neuroscience, Physiology and Pharmacology, University College London, London, United Kingdom

**Abstract** Adapting neural representation to rapidly changing behavioural demands is a key challenge for the nervous system. Here, we demonstrate that the output of the primary olfactory area of the mouse, the olfactory bulb, is already a target of dynamic and reproducible modulation. The modulation depends on the stimulus tuning of a given neuron, making olfactory responses more discriminable through selective amplification in a demand-specific way.
DOI: https://doi.org/10.7554/eLife.43558.001

## Introduction

Behavioural contexts often pose conflicting demands on neural representations of stimuli. A representation that is optimal for one behaviour may be unsuitable for another, as seen in the requirements for discriminating between stimuli vs. generalizing over the same stimuli. Adjusting representations transiently may enable the organism to cope with rapidly changing behavioural contexts.

Previous studies indicate that changes in sensory processing accompany behavioural acquisition of olfactory discrimination as early as in the primary olfactory region of the mouse, the olfactory bulb (OB). Some of these changes depend on the reward association (*Doucette and Restrepo, 2008*), which in turn also correlate with changes in stimulus sampling patterns (*Jordan et al., 2018*; *Verhagen et al., 2007*). In addition, recent reports indicate that the nature of the task, too, is an important determinant. For example, as mice learn olfactory discrimination tasks over days, how the OB output changes during this period depends on the difficulty of the task, where the change in response decorrelation depends on the similarity of odours used (*Chu et al., 2016*; *Yamada et al., 2017*). However, some of these changes may be associated with animals becoming familiar with the stimuli over days (*Chu et al., 2016*). Importantly, whether such task-related changes in the OB are dynamic, and how they are implemented remain unclear.

Here, we investigate how behavioural demands shape olfactory responses of OB output as mice switch between tasks that differ in difficulty. We find that olfactory processing in the OB in response to identical stimuli changes rapidly with task demands, in a manner that is suited to the task at hand.

## Results

To study how behavioural demands influence olfactory processing, we used two olfactory discrimination tasks, namely, coarse versus fine discrimination (*Figure 1A*). Olfactory stimuli used in the coarse discrimination task are easily distinguishable, while stimuli employed in the fine discrimination task are more similar (*Figure 1—figure supplement 1*). In both tasks, the rewarded odour α (S+ odour) was a mixture of two odorants (A and B), mixed at a concentration ratio of 40%/60%. The nature of

**\*For correspondence:**
andreas.schaefer@crick.ac.uk
(ATS);
izumi.fukunaga@oist.jp (IF)

**Competing interests:** The authors declare that no competing interests exist.

**Figure 1.** OB output neurons change olfactory representation rapidly and reproducibly with demands. (a) Structure of discrimination tasks. (*Left*) Fine discrimination. A trial starts with two flashes of an LED. A rewarded stimulus (α) is a mixture of A and B odours at a concentration ratio of 40:60% (α-odour), associated with a water (reward). In a non-rewarded trial, the A/B mixture is presented at a concentration ratio of 60:40% (α'-odour), and no reward is given. (*Right*) Coarse discrimination. A trial starts with one flash of an LED. The rewarded odour is the same A/B 40/60 mixture as in fine

*Figure 1 continued on next page*

*Figure 1 continued*

discrimination. A non-rewarded stimulus is a mixture of different odours (β). On some rewarded trials, α' odour is presented to assess whether mice generalize across both A/B mixtures. (**b**) Timeline of experiment. Each session lasted ~20 min, and occurred once a day. *Inset*: Example of switching sessions used for a typical animal (six sessions shown). Variable epoch length ensures that at least four rewarded trials appear per epoch. (**c**) Trial-by-trial average performance across five mice for all trials (**c1**), as well as trials around switching (**c2,c3**). Mean and s.e.m. are shown. (**d**) Imaging configuration. GCaMP6f fluorescence from mitral and tufted cells was imaged with a two-photon microscope through a chronic window (*middle*) in head-fixed mice performing task switching. *Right*, an example field of view. Scale bar = 50 μm. (**e**) Trajectories corresponding to the α and α' odours during fine and coarse discrimination, plotted as trajectories in the first three principal components. Pseudo-population response constructed from all ROIs (n = 353, five mice). (**f**) *Left*, Trial-by-trial α odour response (average amplitude during 1 s odour presentation) for example ROIs. Mean and s.e.m. are shown as horizontal lines. *Middle*, Corresponding transients averaged for each task type. *Right*, The same traces as in the middle panel, but zoomed into the odour period. (**g**) Time course of change among significantly modulated ROIs. The response amplitude for an odour relative to response amplitudes during the first fine discrimination epoch (mean ±s.e.m.; n = 42 ROIs, five mice). The observation holds true when the rewarded trials are not preceded by a non-rewarded trial (data not shown).

DOI: https://doi.org/10.7554/eLife.43558.002

The following figure supplements are available for figure 1:

**Figure supplement 1.** Method for imposing different behavioural task demands.

DOI: https://doi.org/10.7554/eLife.43558.003

**Figure supplement 2.** Task-switch learning and probe trials.

DOI: https://doi.org/10.7554/eLife.43558.004

**Figure supplement 3.** Movement and sniff patterns do not contribute to task-dependent differences.

DOI: https://doi.org/10.7554/eLife.43558.005

**Figure supplement 4.** Task-dependent modulation observed during behaviour is absent under anaesthesia.

DOI: https://doi.org/10.7554/eLife.43558.006

the task depended only on the non-rewarded odour (S- odour). In the coarse discrimination task, the S- odour was odour β, comprising odorants C and D, which are not present in the rewarded stimulus. On the other hand, the S- odour used in the fine discrimination task was odour α', made by mixing odorants A and B, but mixed at a different concentration ratio (60%/40%). By using the rewarded odour, α, in both tasks, that is, by making the odour identity and reward association consistent, we isolate the influence of task demands when investigating neuronal responses.

Following sequential training for coarse and fine discrimination over two weeks, head-fixed mice were trained to switch rapidly between the two tasks within the same imaging session (*Figure 1a,b*). A typical session lasted approximately 20–30 min, consisting of two fine discrimination epochs (designated 'Fine 1' and 'Fine 2') and one coarse discrimination epoch that occurred between the two fine discrimination epochs, with each epoch 15–25 trials long. This design was used to control for time-dependent effects. On some trials during the coarse discrimination, odour α' (the 60A/40B mixture) was presented as a rewarded odour ('probe' trials). This modification forced mice to generalize over A/B mixtures (α and α') during coarse discrimination, and to discriminate between the α and α' odours during fine discrimination. Since our focus is to study how the α-odour response changes with task demands, probe trials were presented, on average, 1.6 times per session only. Over the course of approximately four days, mice learn to perform the task switching with high accuracy (*Figure 1c*, *Figure 1—figure supplement 2*).

To assess the effect of task demands on OB processing, olfactory responses of the principal neurons, mitral and tufted cells (M/TCs), were imaged in trained mice during behaviour using a two-photon microscope (*Figure 1d*). A genetically encoded calcium indicator, GCaMP6f (*Chen et al., 2013*), was expressed conditionally in M/TCs in the OB by crossing the Ai95D mouse line (*Madisen et al., 2015*) with the Tbet-ires-Cre mouse line (*Haddad et al., 2013*). Imaging was accomplished through a previously implanted chronic imaging window (*Holtmaat et al., 2009*).

To visualize if and how M/TC responses as a whole are modulated by tasks, calcium transients from all regions of interest (ROIs; 353 ROIs from five mice) were expressed as pseudo-population vectors and plotted as trajectories in the first three principal components (*Figure 1e*). Despite the fact that the odour is identical, the trajectory for the α odour during coarse discrimination lies distinctly away from that for fine discrimination. On the other hand, the trajectories for odour α from the first and second epochs of fine discrimination superimpose closely. This indicates that olfactory representation in the OB changes with task, but reversibly.

At the level of individual cells, a subset of MTCs was found to change its responses significantly with task (42/353 ROIs; 17/353 ROIs in shuffle control show significant change). Single-trial analysis revealed that lick and sniff patterns do not explain this task-related change (*Figure 1—figure supplement 3*). In fact, when variability arising from sniff patterns was removed through linear regression, a greater proportion of ROIs were found to be significantly modulated by task (56/353 ROIs; *Figure 1—figure supplement 3*). In addition, this task-related modulation is absent in mice anaesthetized with ketamine and xylazine (*Figure 1—figure supplement 4*). Importantly, the change is present immediately after task switching (*Figure 1f,g*). Among the task modulated ROIs, significant change is observed even in the first trial after switching from fine to coarse discrimination (mean change in the α odour response during $1^{st}$ coarse trial = $-0.17 \pm 0.04$ ΔF/F; p<0.01, t-test for equal means, t-score = $-4.3$, n = 43 ROIs, five mice). Switching back to fine discrimination, responses become comparable to the original amplitudes by the $3^{rd}$ trial (mean difference in α odour response relative to Fine 1 = $-0.05 \pm 0.04$ ΔF/F; p=0.26, t-score = $-1.2$), closely mirroring the recovery time course of behavioural accuracy (*Figure 1c*).

Overall, M/TCs tend to increase their responses during fine discrimination (*Figure 2a,b*). The increase is particularly pronounced when mice show clear evidence of switching, as assessed by the performance during probe trials (*Figure 2—figure supplement 1*).

According to a previous, longitudinal study, when mice learn to perform fine discrimination, there is an accompanying increase in the fraction of divergent responses among M/TCs (*Chu et al., 2016*). Accordingly, we assessed whether stimulus selectivity is an important parameter in the dynamic, task-related change observed here. To this end, for each ROI, we measured its selectivity to α vs α' odours using the t-score (or t-statistic), which compares mean response amplitudes (see Methods). Intriguingly, we found that it is the α -selective M/TCs that enhance their responses during fine discrimination (*Figure 2c–e*; mean change = $0.06 \pm 0.02$ ΔF/F for alpha selective ROIs; p = 0.008, paired t-test for equal means between α-selective and α'-selective ROIs; n = 21 sessions, five mice).

As a result, stimulus-selective neurons are over-represented among task-modulated M/TCs (*Figure 2f*; p = 0.01, 2-sample K-S test for equal distributions, K-S statistic = 0.33, n = 22 task-modulated ROIs and 201 non-modulated ROIs from sessions with probe trials). Notably, 24% of stimulus-selective neurons were significantly task modulated, compared to 10% of the entire population of neurons, together suggesting a critical role that modulation plays in discriminating similar stimuli. Consistently, when these task-modulated ROIs are removed from the analysis, the population of M/TC responses to α and α' odours became more correlated (*Figure 2g*; mean % change in Pearson's Correlation = $2 \pm 1$ when modulated ROIs were removed, and $0.2 \pm 0.1$ when random ROIs were removed; p=0.04, paired t-test, n = 5 mice) and the decoder performance deteriorated specifically for fine discrimination (*Figure 2—figure supplement 2*). We note that a general increase in response amplitudes alone does not explain our result, as an overall increase in response magnitude in an anaesthetized preparation does not lead to enhanced stimulus selectivity and discriminability (*Figure 2—figure supplement 3*).

## Discussion

Overall, we find that rapid, task-dependent modulation of odour responses in the primary olfactory area occurs dynamically to enhance odour representation to suit the behaviour. We find that modulation occurs even when the stimulus-reward association is identical, isolating context as the only variable. Concomitantly, discriminability of stimulus representations is altered through selective boosting of responses of discriminating neurons (*Figure 2h*). These divergent responses occurred in about 12% of the neurons we observed, closely matching a previous report (*Chu et al., 2016*).

Similarly, task-related modulation is not easily explained by the difference in the stimulus statistics (frequency of A/B mixture presentations), as the modulation is absent in mice anaesthetized with ketamine and xylazine (*Figure 1—figure supplement 4*), although we cannot exclude that the influence of stimulus-statistic might be state-dependent. How, then, might dynamic and selective modulation arise in the OB? Unusually for a primary sensory area, the OB receives dense feedback projections from olfactory cortices (*Boyd et al., 2012*; *Davis and Macrides, 1981*; *Markopoulos et al., 2012*; *Otazu et al., 2015*; *Rothermel and Wachowiak, 2014*) as well as neuromodulatory centres (*Macrides et al., 1981*; *McLean and Shipley, 1987*; *Steinfeld et al., 2015*). Among these, possible pathways for selective amplification include direct excitatory feedback, such

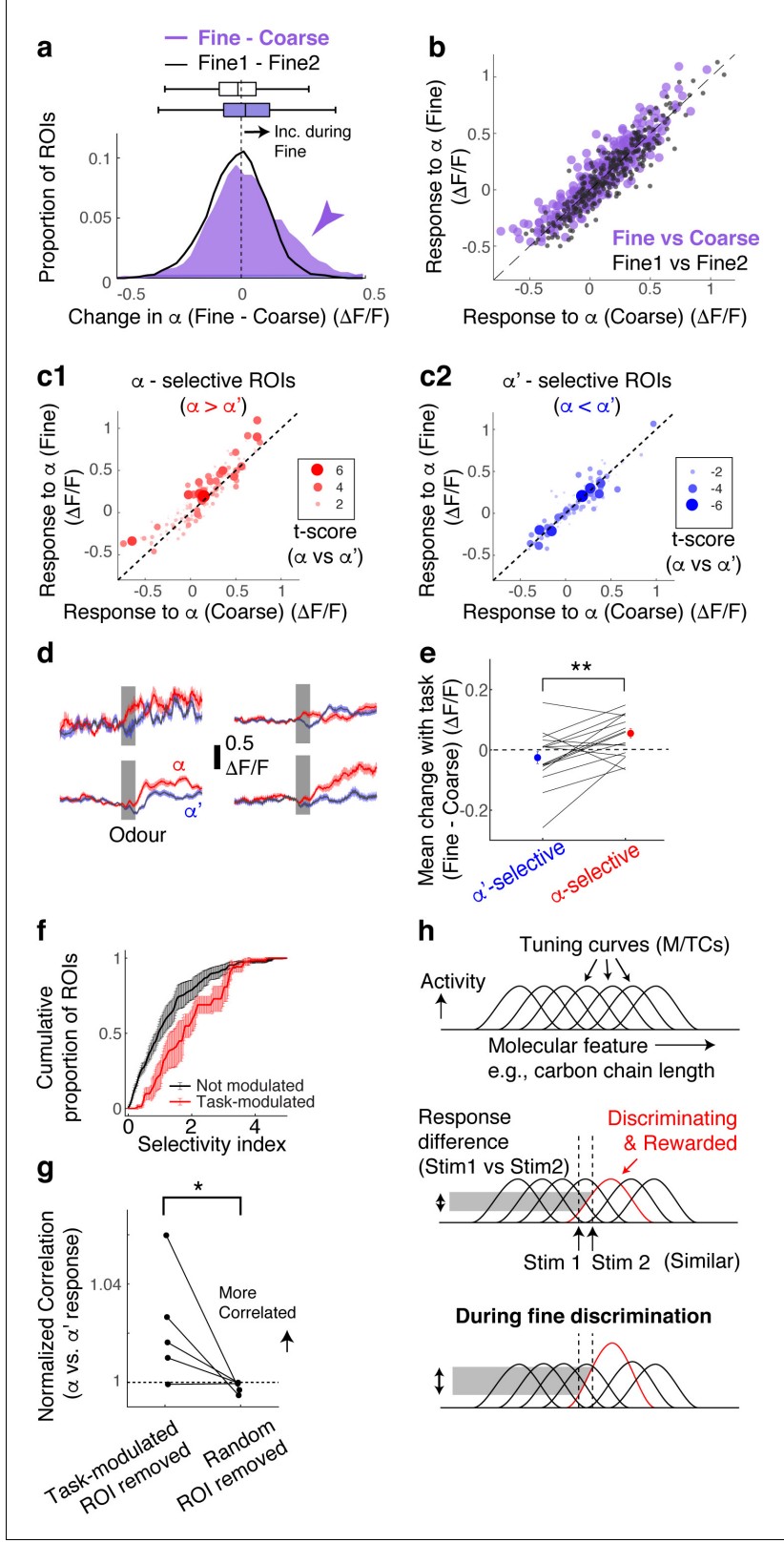

**Figure 2.** Selective amplification leads to demand-specific enhancement of stimulus decorrelation. (a) Histogram of task-dependent change in α responses (response during fine discrimination – response during coarse discrimination; purple). Within-task variability (black line) is shown for comparison. (b) Comparison of odour responses during coarse (x-axis) and fine (y-axis) discrimination for all ROIs (purple). Within-task comparison (Fine1
*Figure 2 continued on next page*

*Figure 2 continued*

vs Fine2; black points) shown for reference. (**c**) Task-dependent odour response comparison as in (**b**), but separated by odour selectivity. α-selective ROIs (red) are ROIs with t-scores greater than zero, where the strength of the selectivity is indicated by the size of the marker. α'- selective ROIs are those with t-scores <0. (**d**) Average transients in response to odour α (red) and odour α' (blue) from 4 ROIs selective for odour α. Error bar = s.e.m. (**e**) Summary of task-dependent change for odour α for ROIs grouped by selectivity. Each data point = average for each imaged location. N = 21 sessions, five mice. Sessions where probe trials were omitted are excluded from this analysis. (**f**) Distribution of selectivity indices for task-modulated ROIs (red; n = 32 ROIs, five mice) and ROIs without task modulation (black; n = 255 ROIs, five mice). The selectivity index is the absolute value of the t-score for α-responses vs α'-responses imaged during fine discrimination. (**g**) Correlation coefficients between α vs α' responses, where a subset of ROIs is removed and normalized to when a full set of ROIs used. Each coefficient is from ROIs from individual animals, instead of imaging location, due to a small number of ROIs in some imaged locations (n = 5 mice). (**h**) Schematic of finding. MTCs tuned to different 'molecular features,' for example, distinct carbon chain lengths, have different tuning curves along this axis. Some MTCs respond differently to two similar stimuli (stim 1 and stim 2), which contribute the most to discrimination. MTCs that are selectively tuned to the rewarded odour are enhanced when mice need to discriminate between the two similar odours, resulting in enhanced discriminability.

DOI: https://doi.org/10.7554/eLife.43558.007

The following figure supplements are available for figure 2:

**Figure supplement 1.** Task-related change is prevalent in sessions with clear evidence of switching.
DOI: https://doi.org/10.7554/eLife.43558.008
**Figure supplement 2.** Role of task-modulated neurons in task-dependent stimulus discrimination.
DOI: https://doi.org/10.7554/eLife.43558.009
**Figure supplement 3.** Responses are more reliably discriminated in behaving mice.
DOI: https://doi.org/10.7554/eLife.43558.010
**Figure supplement 4.** PID signals show stable odour pulses.
DOI: https://doi.org/10.7554/eLife.43558.011
**Figure supplement 5.** Examples of imaged planes.
DOI: https://doi.org/10.7554/eLife.43558.012
**Figure supplement 6.** Estimation of contribution from out-of-focus fluorescence.
DOI: https://doi.org/10.7554/eLife.43558.013

---

as a direct excitatory drive from centrifugal inputs onto M/TCs, as described for the AON (*Markopoulos et al., 2012*), or modulation of inhibitory neurons via some competitive mechanism (*Koulakov and Rinberg, 2011*), although we cannot rule out a role for neuromodulators. So far, there is no evidence that olfactory signals carried by the cortical feedback are spatially matched to local (glomerular) olfactory representations (*Boyd et al., 2015*). Whether cortical feedback to the OB indeed forms a pattern akin to an 'attention-field' to enhance relevant signals (*Reynolds and Heeger, 2009*) will be a key topic of future investigation.

Due to the limitations from kinetics and sensitivity of the genetically encoded calcium indicators (*Chen et al., 2013*), we chose to analyse the 1 s window during odour presentations. This calcium response could therefore potentially reflect a variety of parameters related to behaviour, in addition to a purely sensory component. Indeed, previous experiments demonstrated that mice take less than 1 s to arrive at decisions in similar tasks (*Abraham et al., 2004*; *Rinberg et al., 2006*; *Uchida and Mainen, 2003*). It is therefore not inconceivable that some of the task-dependent differences might reflect aspects of decision variables.

According to models of attention developed especially in the primate visual system, the extent of modulation depends on many factors, including the similarity of the attended feature and the neurons' tuning, the exact stimuli used, as well as the stages of sensory processing along a hierarchy (reviewed in *Reynolds and Heeger, 2009*). Indeed, a variety of modulation sizes have been observed at different levels of visual processing: 8% in V1, 26% in V4 (*McAdams and Maunsell, 1999*; *Moran and Desimone, 1985*; *Motter, 1993*), 19–60% in MT and 40% in MST cells (*Treue and Maunsell, 1996*; *Treue and Martínez Trujillo, 1999*). In discriminating similar stimuli, neurons that contribute the most are those that respond differently to the stimuli used. Modelling studies suggest that these are most often neurons with a preferred stimulus feature that is shifted slightly away from the stimulus features to be discriminated (*Jazayeri and Movshon, 2006*). These, in our case,

comprise about 12% of the imaged neurons, which closely match previous reports (*Chu et al., 2016*). Our results suggest that such neurons may be the target of selective and dynamic modulation to enable enhanced discriminability as needed. The ability to induce different task demands and observe the resultant modulation in a primary sensory area may prove useful to understand how the brain implements solutions that meet ever changing behavioural demands.

## Materials and methods

### Animals/surgery

All animal experiments were approved by the institutional veterinarian and ethics committee of OIST Graduate University (2016–3) and have been performed in strict accordance. All recovery surgery was carried out using standard aseptic technique under isoflurane anaesthesia and carprofen analgesia. Adult male mice (6–8 weeks old) from the cross between the Tbet-Cre line (*Haddad et al., 2013*) and the Rosa26-GCaMP6f line (*Madisen et al., 2010*) were used. For chronic optical access, a method similar to that of *Holtmaat et al. (2009)* was used. Briefly, a small, round piece of glass (~1 mm diameter) was cut from a cover slip (borosilicate glass 1.0 thickness) using a diamond knife (Sigma-Aldrich). It was fitted over a craniotomy made above the left olfactory bulb and sealed with cyanoacrylate (Histoacryl, TissueSeal, USA). A custom-made head-plate was attached to the exposed, dried occipital bone using gel superglue and dental cement. Non-steroidal anti-inflammatory treatments (Carprofen, i.p.) were given for 3 days post-operatively. All behavioural sessions began more than 2 weeks after the surgery.

### Olfactometry

Odours were presented using a custom-made flow-dilution olfactometer similar to an earlier design (*Fukunaga et al., 2012*), except in the control of odour concentrations (*Figure 2—figure supplement 4*). Odour concentration was set using precise, pulsatile packets of saturated odour into a 50 mL flask, where it was mixed with a steady flow of background air (one slpm) in order to eliminate fast transients in odour concentration (time constant =~ 400 ms). The final odour concentration presented to the animal was approximately 0.5–1% of the saturation level. A binary mixture was generated by mixing two streams of odorized air in the same mixing compartment, and concentrations were verified using a photoionization detector for each odour component (where the tested odour was mixed with a blank control that did not give signals; *Figure 2—figure supplement 4*). Each odour canister had a large headspace (~40 mL) over the odorants to minimise run-down over time. Teflon tubing and air purge (~5 slpm for 10 s) during the inter-trial interval were used to minimize contamination.

### Go-No-Go olfactory discrimination

Water access in the home cage was restricted, and head-fixed mice went through habituation sessions before discrimination training commenced. During habituation sessions, animals were head-fixed and were presented water from the port. This stage of training lasted 15–20 min a day, until animals were comfortable enough to drink water from the water port. This was typically achieved within two days. Animals were then trained to associate one olfactory stimulus with a water reward (S+ odour) and another olfactory stimulus with no reward (S- odour). The correct response for the S+ trials was to lick the water port for a reward during 2 s after the onset of odour presentation, while the correct response for S- trials was to refrain from licking. The reward was a single drop of water approximately 20 uL in volume. The water port served also as a lick sensor (beam break; PM-F25, Panasonic, Japan), which was coated with black silicone externally to prevent light leakage. Respiratory rhythms were monitored continuously through the contralateral naris using a fast mass flow sensor (FBAM200DU, Sensortechnics, Germany). Odours used were ethyl butyrate and eugenol in the rewarded odour mixture, and methyl salicylate and methyl tiglate in the non-rewarded odour mixture used in the coarse discrimination. The sequence of S+ and S- trials was chosen so that no more than three consecutive trials were of the same type, but otherwise random. On each trial, odour was presented for 1 s, and inter-trial interval was 20 s. Animals were monitored closely and at no point did animal's weight decrease below 80% of the initial weight.

## Imaging

Data in this manuscript were obtained from awake, head-fixed mice engaged in the tasks, as well as anesthetized mice, as indicated, through previously implanted optical windows. Imaging in awake animals took place after approximately 4 days of task-switch training, when mice had acquired the task-switching. In these trained mice, imaging sessions took place for up to 6 days depending on the quality of the imaging window. Each session lasted, on average, for 30 min. The exact number of trials mice performed each day varied depending on the level of motivation. However, to balance time-dependent effects, including bleaching, the same number of trials from the first and second fine discrimination epochs were analysed. That is, the later trials during the $2^{nd}$ epoch of fine discrimination was truncated from analyses. Imaging in anaesthetized mice took place after the switching sessions. Two-photon fluorescence of GCaMP6f was measured with a custom-fitted microscope (INSS, UK) fitted with a 25x Objective (Nikon N25X-APO-MP1300, 1.1 N.A.), and high-power laser (930 nm; Insight DeepSee, MaiTai HP, Spectra-Physics, USA) at depths 50–400 µm below the surface of the olfactory bulb. Each field view was 256 µm x 256 µm (512 × 512 pixels). Images from a single plane were obtained at ~30 Hz with a resonant scanner. Each day, the stage co-ordinate was zeroed at a reference location, guided by the surface blood vessel pattern using an epifluorescence camera. Imaging location was then chosen as an area not overlapping with previous sessions (based on the x, y and z co-ordinates and field view, and confirmed post-hoc by eye; see *Figure 2*; *Figure 2—figure supplement 5*). Modulated cells were found at all depth of imaging, suggesting that both MCs and TCs are subject to task-dependent modulation.

## Analysis

Images were analysed using custom-written routines in Spike2 (Cambridge Electronic Design, UK), Matlab (Mathworks, USA) and macros in Fiji (ImageJ), run on a PC (32 GB RAM, 10 × 2.2 GHz Intel Xeon processors).

### ROI detection

From an average of ~1000 frames for each imaging plane, somata were manually delineated using an ROI manager (ImageJ). Selected oval regions were confined within the perimeter of the soma, and somata with filled nuclei were excluded. Based on the method by *Kerlin et al. (2010)* (*Figure 2—figure supplement 6*), the contribution from out-of-focus fluorescence was approximately 60%.

### Odour response amplitude

Unless otherwise stated, the average odour response was obtained from ΔF/F values during the first 1 s of odour presentation (after the onset of valve opening). Results are qualitatively similar when measurements are from shorter time windows, but noisier, especially for single-trial analyses.

### Similarity of S+ vs S- representations

For each pair of responses per imaging location, the relative fluorescence change (ΔF/F values) for all ROIs in the field of view was treated as vectors and Pearson correlation coefficient was calculated.

### Odour response

Significant responses were defined as those in which the magnitude of odour-evoked fluorescence change exceeded 3x the standard deviation of baseline fluctuations.

### Task-dependent change

To determine if a task-dependent change was statistically significant, evoked response amplitudes for S+ trials for fine and coarse discriminations were tested for the same distributions (two-sample t-test for no difference, two tailed).

## Principal component analysis

The method is based on *Niessing and Friedrich (2010)*. Briefly, for each ROI, GCaMP6f transients (ΔF/F values) from each condition (odour and task) were concatenated and principal components obtained using the Matlab function *pca*. Original data were projected on the new components (first three principal components) to obtain trajectories.

## Stimulus selectivity (T-score)

For each ROI, odour response amplitudes were obtained from α trials and α' trials. These were compared using the Matlab function for the two means, namely, the two-sampled t-test (ttest2) to obtain a t-score (t-statistic). The T-score was used instead of the commonly used z-score because of the small number of trials.

## Stimulus selectivity

Selectivity index was defined as the magnitude (absolute value) of the t-score.

## ROI removal and resultant change in stimulus correlation

For each animal, average odour response from ROIs (all imaging sessions) were expressed as one vector. From this, task-modulated ROIs were removed and Pearson's correlation coefficient for α- and α'-responses was calculated. This value was compared against the correlation coefficient obtained from the full set i.e., change in correlation = coefficient (ROI removed) − coefficient (full set))/coefficient (full set). For removing random sets of ROIs, a random subset, comprising the same number of ROIs as the task-modulated set, was selected using the randperm function in Matlab. Then the change in correlation was calculated as above.

## Linear decoder analysis

Matlab function fitcdiscr was used to fit a linear classifier. For each dataset (each imaging session), training data comprised the population of M/TC responses from 2/3 of the trials during the fine discrimination block. The other 1/3 of the trials was used as a test set to predict the label (S+ vs S- odour). % accuracy is the % of trials from the test set that matched the correct label.

## Behavioural performance

Animal lick responses were analysed. For S+ and S- trials, correct responses were the presence and absence of licking during a window 1–2.5 s after odour onset, respectively. Incorrect responses were the absence and presence of lick responses for S+ and S- trials in the same window, respectively. The number of correct trials was expressed as the % of all trials.

## Acknowledgements

The authors thank Martyn Stopps for help with the olfactometer design, Charly Rousseau and Nicholas Burczyk for technical advice, Molly Strom and Troy Margrie for reagents, Taha Soliman for technical assistance, OIST mechanical workshop for the custom parts, and Kevin Franks, Alexander Fleischmann, Rebecca Jordan, Cary Zhang, Sander Lindeman, and Steven Aird for helpful comments on the manuscript. This work was supported by Okinawa Institute of Science and Technology Graduate University, the Francis Crick Institute which receives its core funding from Cancer Research UK (FC001153), the UK Medical Research Council (FC001153), and the Wellcome Trust (FC001153); by the UK Medical Research Council (grant reference MC_UP_1202/5); and by the DFG (SPP 1392). AS is a Wellcome Trust Investigator (110174/Z/15/Z).

## Additional information

### Funding

| Funder | Grant reference number | Author |
| --- | --- | --- |
| Okinawa Institute of Science and Technology Graduate University | | Anzhelika Koldaeva Izumi Fukunaga |
| Francis Crick Institute | FC001153 | Andreas T Schaefer Izumi Fukunaga |
| Wellcome Trust | 110174/Z/15/Z | Andreas T Schaefer |
| Medical Research Council | MC_UP_1202/5 | Andreas T Schaefer Izumi Fukunaga |
| Deutsche Forschungsge- meinschaft | SPP 1392 | Andreas T Schaefer |

The funders had no role in study design, data collection and interpretation, or the decision to submit the work for publication.

### Author contributions

Anzhelika Koldaeva, Investigation, Writing—review and editing; Andreas T Schaefer, Conceptualization, Supervision, Funding acquisition, Writing—original draft, Writing—review and editing; Izumi Fukunaga, Conceptualization, Data curation, Formal analysis, Supervision, Investigation, Methodology, Writing—original draft, Writing—review and editing

### Author ORCIDs

Andreas T Schaefer http://orcid.org/0000-0002-4677-8788
Izumi Fukunaga http://orcid.org/0000-0003-1860-5377

### Ethics

Animal experimentation: All animal experiments were approved by the institutional veterinarian and ethics committee of OIST Graduate University (2016-3) and UK Home Office and the Animals and Scientific Procedures Act (#PPL 70/7827), and have been performed in strict accordance.

### Decision letter and Author response

Decision letter https://doi.org/10.7554/eLife.43558.016
Author response https://doi.org/10.7554/eLife.43558.017

## Additional files

### Supplementary files

• Transparent reporting form
DOI: https://doi.org/10.7554/eLife.43558.014

### Data availability

All data generated or analysed during this study are included in the manuscript and supporting files.

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
