## [Decision Letter]

Thank you for submitting your article "Rapid task-dependent tuning of the mouse olfactory bulb" for consideration by *eLife*. Your article has been reviewed by three peer reviewers, including Naoshige Uchida as the Reviewing Editor and Reviewer #1, and the evaluation has been overseen by Gary Westbrook as the Senior Editor.

The reviewers have discussed the reviews with one another and the Reviewing Editor has drafted this decision to help you prepare a revised submission.

Summary

In this short report, Koldaeva and colleagues examined how task demands affect odor representations in the olfactory bulb (OB). Mice performed alternating blocks of fine and coarse odor discriminations. In fine discrimination blocks, mice discriminated two odor mixtures (α versus α'; 40%/60% versus 60%/40% binary mixtures of odor A and B). In coarse discrimination blocks, mice discriminated two odor mixtures comprising different odorants (α versus β; a 40%/60% mixtures of A and B versus a 40%/60% mixture of C and D). In all conditions, odor α is associated with reward (S+) whereas α' or β is associated with no outcome (S-). The authors performed two-photon calcium imaging to monitor the activity of mitral/tufted cells (M/TCs) in the OB. The authors found that some M/TCs that responded to odor α increased their response to this odor in the fine compared to coarse discrimination block. These M/TCs did not change their responses to odor α' (probe trials). Based on decoding analysis, the authors show that increased responses are accompanied better discrimination of odors. These results indicate that odor evoked responses of M/TCs rapidly change to reflect a task demand.

All the reviewers thought that this work addresses an important and timely question. The rapid change in odor response is striking. The manuscript is written clearly and the data are generally convincing. However, the reviewers' opinions on overall significance and novelty varied. One reviewer raised concerns on novelty of the work over a previous work (Doucette and Restrepo, 2008) as well as on the effect size of neuronal changes. After discussion, we agreed that these concerns could be addressed by clarification in text and additional analysis. If the authors can fully address these points, we are happy to consider a revised manuscript.

Essential revisions:

1) One reviewer raised the issue of novelty. Specifically, a task-dependent modulation of M/TC activity was documented in Doucette and Restrepo, 2008. The authors should reference and discuss this work. It is critical that the authors clarify the novelty of the present work over the Doucette study. One potentially novel observation is that only M/TCs are sensitive to fine versus coarse discrimination. However, we thought that this point should be better demonstrated and it should be properly discussed.

2) The effect size of the modulation is generally small. One reviewer suggested that the authors could potentially make a case that a small change across a large fraction of neurons could have a potentially large effect, or that the most influential signals are buried in their responses somewhere. This was touched upon in the discussion in reference to Jazayeri and Movshon, 2006, as well as Figure 2H. However, the authors could expand this discourse, further contextualizing and supporting the significance of their findings by discussing additional relevant attention literature and comparison to alternative models of sensory selection.

3) The reviewers thought that the current set of experiments could not unambiguously establish behavioural relevance of the observed phenomenon. However, the existing data can be used to substantiate this point. The authors discovered that task-dependent changes are most prominent in cells that might convey more information for fine discrimination, suggesting their relevance in this task. This is an important observation, but this is not thoroughly supported by the reported analyses. A classification analysis would help, where one can see that modulation aids a linear classifier in both the fine and coarse conditions (or that the cells you expect aid performance when included in the classification). Although the authors compare linear decoder performance between the anesthetized and awake state, the benefits of including the significantly modulated population was not explicitly tested. Also, a trial by trial analysis linking trials with more modulation to better performance would strongly favor their hypothesis that these changes are meaningful. Since we don't know when the spikes are occurring, these changes could be happening well after the animal has made their decision and reflect choice or motor-based signals feeding back to the olfactory bulb (not attentional modulation signals per se). All these issues should be thoroughly discussed in paper.

---

## [Author Response]

Essential revisions:1) One reviewer raised the issue of novelty. Specifically, a task-dependent modulation of M/TC activity was documented in Doucette and Restrepo, 2008. The authors should reference and discuss this work. It is critical that the authors clarify the novelty of the present work over the Doucette study. One potentially novel observation is that only M/TCs are sensitive to fine versus coarse discrimination. However, we thought that this point should be better demonstrated and it should be properly discussed.

We thank the reviewers for raising this important point.

The study by Doucette and Restrepo was indeed one of the first to characterize how individual OB neurons change as animals acquire an olfactory discrimination task. They observed transient changes in the firing pattern of these neurons, where the nature of the change depends on the reward association. This reward-associated firing rate change caused neurons to become more divergent in some blocks. This observation has, together with earlier observations by Kay and Laurent (1999) and by Rinberg et al., 2006, since then, inspired a number of studies aimed at understanding the nature of dynamic OB responses, including the effect of sampling strategy (Verhagen et al., 2007; Jordan 2018ab, Shusterman, 2018), general behavioural state (Kato et al., 2012), or the influences of neuromodulator and feedback signals (Petzold et al., 2009, Otazu et al., 2015, Rothermel et al., 2014).

Of the various factors that could potentially influence the OB dynamically, the focus of our current manuscript is, as the reviewers kindly point out, on the task demand. We believe the manuscript differs from previous works that investigated the influence of task demands as no previous work explicitly tested if the demand-specific modulation of the OB is indeed dynamic. Specifically, we asked whether the task demand on its own modulates the OB – independent of other parameters such as response, satiation, motivation and reward association. This was the motivation for developing our novel behavioural paradigm, where mice switch rapidly between coarse and fine discrimination. With this task design, we managed to keep the above parameters intentionally constant. In addition, we also show that other potentially confounding factors such as sampling behaviour remain constant as well, isolating the dynamically changing task demand as the key variable.

In summary, the study by Doucette and Restrepo played an important role in the study of the OB as a dynamic, active filter and therefore should have been cited. We sincerely apologise for this omission. We have now modified the Introduction substantially to acknowledge the study, as well as to explain the emphasis of our paper (nature of the task) more clearly. We added the following text:

“Previous studies indicate that changes in sensory processing accompany behavioural acquisition of olfactory discrimination as early as in the primary olfactory region of the mouse, the olfactory bulb (OB). Some of these changes depend on the reward association (Doucette and Restrepo, 2008), which in turn can also correlate with changes in stimulus sampling patterns (Jordan et al., 2018; Verhagen et al., 2007). In addition, recent reports indicate that the nature of the task, too, is an important determinant. For example, as mice learn olfactory discrimination tasks over days, how the OB output changes during this period depends on the difficulty of the task, where the decorrelation of stimulus representations depends on the similarity of odors used (Chu et al., 2016; Yamada et al., 2017). However, some of these changes may be associated with animals becoming familiar with the stimuli over days (Chu et al., 2016). Importantly, whether such task-related changes in the OB are dynamic, and how they are implemented remain unclear.”

Furthermore, regarding our novel observation, the manuscript now contains analysis and discussion in greater detail, elaborating on the observation that, the MCs/TCs that are modulated by task demand are those that differentially respond to the similar odour stimuli. For detail, please see our response to major point 2 below.

2) The effect size of the modulation is generally small. One reviewer suggested that the authors could potentially make a case that a small change across a large fraction of neurons could have a potentially large effect, or that the most influential signals are buried in their responses somewhere. This was touched upon in the discussion in reference to Jazayeri and Movshon, 2006, as well as Figure 2H. However, the authors could expand this discourse, further contextualizing and supporting the significance of their findings by discussing additional relevant attention literature and comparison to alternative models of sensory selection.

Thank you for this comment and the helpful suggestion to explain the significance our work in context of previous works on sensory selection at large.

As the reviewers point out, on first glance, the effect size might appear relatively small. According to models of attention developed especially in the (primate) visual system, the gain modulation depends on many factors, in particular the similarity of the attended feature and the neurons’ tuning (reviewed in Reynolds and Heeger, 2009), as well as the stages of sensory processing along a hierarchy. Indeed, a variety of modulation sizes have been observed at different levels of visual processing: 8% in V1, 26% in V4 (McAdams and Maunsell, 1999), 19% in MT and 40% in MST cells (Treue and Maunsell, 1996). It must be noted that these studies used stimuli that are thought to best drive the neurons (stimuli presented to the receptive field and the stimulus orientations matched to the tuning of the recorded cells).

If we similarly focus our analysis on cells that are most informative for the task at hand (cells that are selective for α odor, responding differently to the 40/60 and 60/40 mixtures), we note that a substantial fraction of those (24%) is significantly modulated by task demand (Author response image 1). Thus, the seemingly small effect size may partially be explained by our somewhat conservative analysis method that includes all cells, irrespective of their stimulus tuning properties.

**Author response image 1. respfig1:** Histogram of task-dependent change, expressed as t-score (fine vs. coarse). Overall, 10% of neurons change with task. Among the α-selective ROIs (cells responding differently to the 40/60 and 60/40 mixtures) this proportion is substantially higher (25%, red). These α-selective ROIs make up approximately 12% of all ROIs – closely matching published results for divergent cells in a long learning task studied by Chu et al., 2016.

Importantly, however, we focus our discussion on the impact of these changes – how does the modulation of this fraction of neurons affect the performance of stimulus discrimination? Here, our new analysis (suggested by the reviewers below) demonstrates that these selective neurons indeed play a significant role in the classifier performance (Figure 2—figure supplement 2). Altogether, our results demonstrate not only the consistency with respect to other attentional studies, but also the importance of modulating the key population of neurons.

To explain this, we have included Figure 2—figure supplement 2 and have modified our Discussion in two locations, which now reads as follows

“Overall, we find that rapid, task-dependent modulation of odor responses in the primary olfactory area occurs dynamically to enhance odor representation to suit the behaviour. We find that modulation occurs even when the stimulus-reward association is identical, isolating context as the only variable. Concomitantly, discriminability of stimulus representations is altered through selective boosting of responses of discriminating neurons (Figure 2H). These divergent responses occurred in about 12% of the neurons we observed, closely matching a previous report (Chu et al., 2016).”

“As a result, stimulus-selective neurons are over-represented among task-modulated M/TCs (Figure 2F; p = 0.01, 2-sample K-S test for equal distributions, K-S statistic = 0.33, n = 22 task-modulated ROIs and 201 non-modulated ROIs from sessions with probe trials). Notably 24% of stimulus-selective neurons were significantly task modulated, compared to 10% of the entire population of neurons, together suggesting a critical role that modulation plays in discriminating similar stimuli.”

“According to models of attention developed especially in the primate visual system, the extent of modulation depends on many factors, including the similarity of the attended feature and the neurons’ tuning, the exact stimuli used, as well as the stages of sensory processing along a hierarchy (reviewed in Reynolds and Heeger, 2009). […] Our results suggest that such neurons may be the target of selective and dynamic modulation to enable enhanced discriminability as needed. The ability to induce different task demands and observe the resultant modulation in a primary sensory area may prove useful to understand how the brain implements solutions that meet ever changing behavioural demands.”

3) The reviewers thought that the current set of experiments could not unambiguously establish behavioural relevance of the observed phenomenon. However, the existing data can be used to substantiate this point. The authors discovered that task-dependent changes are most prominent in cells that might convey more information for fine discrimination, suggesting their relevance in this task. This is an important observation, but this is not thoroughly supported by the reported analyses. A classification analysis would help, where one can see that modulation aids a linear classifier in both the fine and coarse conditions (or that the cells you expect aid performance when included in the classification). Although the authors compare linear decoder performance between the anesthetized and awake state, the benefits of including the significantly modulated population was not explicitly tested. Also, a trial by trial analysis linking trials with more modulation to better performance would strongly favor their hypothesis that these changes are meaningful. Since we don't know when the spikes are occurring, these changes could be happening well after the animal has made their decision and reflect choice or motor-based signals feeding back to the olfactory bulb (not attentional modulation signals per se). All these issues should be thoroughly discussed in paper.

Thank you for this very helpful comment. We agree that further analyses in this direction would improve the manuscript. We implemented your suggestions as follows.

The classifier performance with and without the modulated neurons indeed provided a useful indication of these neurons’ importance. This is the analysis presented in Figure 2—figure supplement 2. Without the task-modulated neurons, the performance of the classifier was significantly degraded for the fine discrimination, while the classifier performance for coarse discrimination was not affected. This demonstrates that, despite the small size of modulated neurons, they play a key role in the discriminability of representations during fine discrimination.

Since this is central to this manuscript, we have now included this analysis as Figure 2—figure supplement 2 and referred in the main manuscript as follows:

“As a result, stimulus-selective neurons are over-represented among task-modulated M/TCs […], suggesting a critical role that modulation plays in discriminating similar stimuli. Consistently, when these, task-modulated ROIs are removed, the population of M/TC responses to α and α’ odors became more correlated… and the performance of a linear classifier deteriorated specifically for fine discrimination (Figure 2—figure supplement 2)”.

In addition, following reviewers’ suggestion, we also analysed the trial-to-trial relationship between the level of modulation and the behavioural performance (Autor response image 2). Since relevant trials are the rewarded trials (the modulation is on α-odour responses), we assessed if confidence in licking response correlated with modulation. We used the amount of early licking as a proxy for “confidence” (more licks during the early period as proxy – albeit rough one – for more confident decisions). To this end, for each rewarded trial from fine discrimination epochs, we averaged the depth of modulation across all modulated ROIs, and separated the trials into two groups depending on the average depth of modulation (i.e., more vs. less modulated trial groups). This analysis revealed that there was a tendency for lick behaviour to be more confident (more licks during early phase) for more modulated trials.

**Author response image 2. respfig2:** Comparison of licking behaviour for trials with larger modulation vs trials with less modulation. For each ROI and for each trial, the odour response amplitude was expressed as a t-score (in comparison against the mean response amplitude during coarse discrimination, taking into account the trial-to-trial variability). For each trial, the average t-score across ROIs was determined. Based on this average t-score, the trials were grouped into high- and low-modulation groups, and the corresponding licking waveforms were extracted (A,B). Mean lick probability was measured for the time window indicated with a rectangle in B, and normalized by the average lick probability for all trials. For control, trial order was randomized but otherwise the same analysis was carried out.

In addition, however, we noticed that licks tend to be stronger and more sustained for trials that appear at the beginning of a session, perhaps reflecting the time-dependent decrease in animals’ motivation (modulated trials are scattered throughout each session). It may be an interesting future investigation to assess the relationship between motivation and extent of modulation in relation to task-dependent modulation. Due to these multiple interacting factors and the number of additional questions arising – peripheral to the present manuscript – we would prefer to not discuss these analyses in depth in the revised paper.

Finally, we agree that for the most part, it is impossible to obtain faithful accounts of spike times with GCaMP imaging (Chen et al., 2013). We now more explicitly discuss the limitation and its consequences on data interpretation as follows:

“Due to the limitations from kinetics and sensitivity of the genetically encoded calcium indicators (Chen et al., 2013), we chose to analyse the 1 second window during odour presentations. This calcium response could therefore potentially reflect a variety of parameters related to behaviour, in addition to a purely sensory component. Indeed, previous experiments demonstrated that mice take less than 1 second to arrive at decisions in similar tasks (Abraham et al., 2004; Rinberg et al., 2006; Uchida and Mainen, 2003). It is therefore not inconceivable that some of the task-dependent differences might reflect aspects of decision variables.”